# Monitoring Hydrothermal Activity Using Major and Trace Elements in Low-Temperature Fumarolic Condensates: The Case of La Soufriere de Guadeloupe Volcano

Manuel Inostroza [1,2,*], Séverine Moune [1,2,3], Roberto Moretti [1,2], Vincent Robert [2,4], Magali Bonifacie [1,2], Elodie Chilin-Eusebe [1,2], Arnaud Burtin [1] and Pierre Burckel [1]

[1] Institut de Physique du Globe de Paris, CNRS, Université de Paris Cité, F75005 Paris, France; severine.moune@uca.fr (S.M.); moretti@ipgp.fr (R.M.); bonifaci@ipgp.fr (M.B.); chilin@ipgp.fr (E.C.-E.); burtin@ipgp.fr (A.B.); burckel@ipgp.fr (P.B.)

[2] Observatoire Volcanologique et Sismologique de Guadeloupe, Institut de Physique du Globe de Paris, F97113 Gourbeyre, France; vincentevan.robert@ird.fr

[3] CNRS, IRD, OPCG Laboratoire Magmas et Volcans, Université Clermont Auvergne, F63000 Clermont-Ferrand, France

[4] IRD, IMAGO LAMA, F98800 Nouméa, France

* Correspondence: manuelinostrozap@gmail.com

**Abstract:** At the hydrothermal stage, volcanoes are affected by internal and external processes that control their fumarolic and eruptive activity. Monitoring hydrothermal activity is challenging given the diverse nature of the processes accounting for deeper magmatic and shallow hydrothermal sources. A better understanding of these processes has commonly been achieved by combining geochemical and geophysical techniques. However, existing geochemical techniques only include the surveillance of major gas components in fumarolic emissions or major ions in cold/thermal springs. This work presents a long-term (2017–2021) surveillance of major and trace elements in fumarolic condensates from the Cratère Sud vent, a low-temperature steam-rich emission from the La Soufriere de Guadeloupe volcano. This fumarole presented a fluctuating performance, offering a unique opportunity to reveal the behavior of major and trace elements, as well as the physicochemical processes affecting magmatic and hydrothermal sources. Time-series analyses allowed for the identification of pH-related chemical fluctuations associated with (1) variable inputs of deep magmatic components at the root of the hydrothermal system, (2) pressurization episodes of the hydrothermal system with increasing fluid–rock interaction, acid gas scrubbing, and vapor scavenging of metals, and (3) the decreased hydrothermal activity, decreasing scrubbing efficiency. Variations in the volatile content (e.g., S, Sb, B, Cl, Bi, Zn, Mo, Br, Cd, Ag, Cu, and Pb), the amount of leached rock-related elements (e.g., Na, Mg, Al, Si, P, K, Ca, Ti, Cr, Mn, Fe, Rb, Sr, Y, Cs, Ba, REEs, and U), and variations in the concentration of Cl and S alone, are postulated as key parameters to monitor volcanic–hydrothermal systems in unrest, such as La Soufriere. Our results demonstrate that monitoring using condensates is a useful geochemical technique, complementing conventional methods, such as "Giggenbach" soda flasks or the so-called Multigas.

**Keywords:** trace elements; hydrothermal systems; La Soufriere de Guadeloupe; volcano monitoring; water–rock interaction

## 1. Introduction

The fumarolic activity is the surficial representation of a degassing magma body at depth, supplying heat and magmatic fluids. At open-conduit volcanoes, where interaction with host rocks and groundwaters is very limited and ephemeral, as well as in peripheral areas surrounding the conduit system, hot ascending fluids can reach the surface at temperatures of as high as 1000 °C. At closed conduit volcanoes, extended fractured pathways drive the interaction with groundwaters, leading to the formation of a volcanic–hydrothermal

system (see [1–4] for an overview). Thus, volcanic–hydrothermal systems (hereafter called hydrothermal systems) are constituted by a heat source (i.e., magma body) and a network of circulating groundwater with recharge and discharge areas (e.g., [5–9]). Hydrothermal systems are then governed by the equilibrium between internal (e.g., the input of heat and magmatic fluids) and external (e.g., variable input of meteoric water, permeability changes) forces, which strongly influence the physicochemical characteristics of fluids discharged at the surface (e.g., [2,10,11]).

Geochemical techniques based on the discrete sampling of vent fluids (by using pre-evacuated ampoules partly filled with an alkaline solution such as NaOH, $Cd(OH)_2$, or $NH_4OH$; [12–14]), as well as on the sensing of the fumarolic plume (via Multigas devices [15,16]), provide valuable information concerning the input of magmatic fluids and their subsequent evolution during their interaction with hydrothermal aquifers and shallow groundwaters (e.g., [17,18]). However, the secondary processes occurring in the hydrothermal reservoir obscure the geochemical signals of deeply derived, magma-related contributions. Depending on their size and the structural setting driving groundwater circulations, some hydrothermal systems may quickly react to external forcing factors. For example, it was recently reported that the significant input of meteoric waters due to heavy rainfalls into shallow hydrothermal systems can decrease the outlet gas temperatures of fumarolic emissions (e.g., [19,20]). Conversely, extended dry seasons seem to decrease scrubbing processes, amplifying the magmatic signal of discharged fluids (e.g., [10,21]). On one hand, changes in the chemistry of fumarolic gases are essential to discriminate between the deep magmatic and shallow hydrothermal contributions and evaluate the nature and deep causes of volcanic unrest phenomena (e.g., [22–26]). On the other hand, they also allow for tracking the onset of episodes of fluid heating and pressure build-up that may lead to the rapid vaporization of fluids and trigger sudden phreatic explosions without any direct magma involvement (e.g., [27,28]). Recognizing the role of secondary hydrothermal processes and external forcing factors on collected gas data is key to the unambiguous interpretations of unrest phenomena occurring at many closed-conduit volcanoes at the hydrothermal stage (e.g., [3,16]).

Magma and hydrothermal reservoirs are also important sources of the trace elements that are emitted during eruptive and quiescent degassing periods (e.g., [29–31]), whose abundances are mainly governed by the outlet gas temperature, chemical composition of fumarolic gases, and magma composition (e.g., [32–35]). Trace elements are primarily emitted in the gaseous phase in high-temperature magmatic systems, although the gas-to-particle conversion can increase the proportion of trace elements in the aerosol phase in fumarolic emissions with lower temperatures and the presence of water vapor (e.g., [35–38]). These elements have been studied in various volcanoes with magmatic and hydrothermal systems using discrete measurements of gas condensates and aerosols (e.g., [25,33,39]). However, few studies have performed continuous condensate measurements to investigate the geochemical behavior of trace elements, as fumarolic/eruptive activity varies. For example, Symonds et al. [33] investigated the chemical evolution of trace elements over 31 years at the Usu volcano (Japan) based on the relative contributions of volatile-behavior elements such as A and B and rock- and incrustations-related particles. This work described the transition from a magmatic dominated to a more degassed and cooled system induced by external factors such as the infiltration of meteoric waters. However, the geochemical behavior of trace elements was better addressed in the groundwaters and thermal springs of active volcanoes such as Bárðarbunga (e.g., [40]) and La Soufriere de Guadeloupe (e.g., [41,42]). Moreover, studies on the chemistry of acid crater lakes (e.g., [26]), natural analogs of condensates, demonstrated that changes in trace element concentrations could be related to volcanic activity.

The knowledge of the physicochemical factors controlling the abundance of trace elements would then be of enormous importance for volcanoes at the hydrothermal stage, where classic proxies used for volcanic monitoring such as $SO_2$ fluxes and $C/S_T$ ratios [16,43] are not totally reliable due to the predominance of $H_2S$ and scrubbing effects,

and where sudden phreatic eruptions could arise. Here, we present the major and trace element concentrations in condensates collected between 2017 and 2021 from the Cratère Sud fumarole (95 °C–111 °C) at La Soufriere de Guadeloupe volcano, one of the most dangerous in the Lesser Antilles volcanic arc, with recent records of phreatic eruptions (e.g., [44,45]). Our dataset encompassed a peak in activity in April 2018 and an amplification of the magmatic signal usually screened by the hydrothermal system in the second half of 2019 [10,11]. This work aims to constrain the chemical behavior of major and trace elements in condensates during fluctuating fumarolic activity and evaluate changes in their concentration based on the influence of internal and external forces. The seismic and geochemical data collected by the Observatoire Volcanologique et Sismologique de la Guadeloupe of the Institut de Physique du Globe de Paris (OVSG-IPGP) are used to investigate the influence of these forces. This work provides valuable information for monitoring low-temperature fumarolic emissions in volcanoes prone to phreatic eruptions and proves that condensates can be used as a prominent monitoring tool, in combination with traditional geochemical and geophysical techniques.

## 2. Geological Background

La Soufriere de Guadeloupe (hereafter La Soufriere) is an andesitic composite volcano located in the Lesser Antilles volcanic arc (16.0446° N, 61.6642° W, 1467 m. a.s.l.; Figure 1a), formed by the subduction of the Atlantic plate below the Caribbean one. La Soufriere was built in the last 11,500 years by recurrent magmatic eruptions and volcanic edifice collapse [45,46]. A good example was the 1530 eruption, triggered by an andesitic magma intrusion that caused the partial collapse of the edifice, followed by a sub-plinian phase culminating with a dome growth, which is currently forming the volcano summit (Figure 1b; [46]). Recently, young fallout deposits found on the north flank of the volcano demonstrated the occurrence of a sub-plinian eruption in 1657 AD [47,48]. After that, La Soufriere experienced frequent phreatic explosions (e.g., 1797–1798, 1836–1837, 1956, 1976–1977), which were restricted to the volcano summit, with the latter leading to the evacuation of 76,000 people from the volcano's surroundings [49]. Later, the volcano remained in post-eruptive quiescence until 1992, when fumarolic and seismic activity reactivated and evolved, as described in Brombach et al. [50], Allard et al. [51], Villemant et al. [41,42], Rosas-Carbajal et al. [52], Tamburello et al. [53], Moretti et al. [11], Jessop et al. [54], and Moune et al. [10].

Fumarolic dry gases discharged from La Soufriere primarily have a magmatic origin [42,50]. However, they evolve due to interactions with the hydrothermal system fed by huge amounts of meteoric waters, which, on the La Soufriere summit, reach up to 10 my$^{-1}$ (5–7 my$^{-1}$ since 2015; [55] and references therein). The main fumarolic emissions (Figure 1c) are in punctual areas, along with fractures (such as Cratère Sud, hereafter "CS") and open pits (such as Gouffre 56 and Tarissan) caused by past phreatic explosions (Figure 1c). Additionally, numerous minor emissions (i.e., pervasive soil degassing) emerge from the ground in an expanding hot zone on the northeast section of the dome (Figure 1c), which includes the Napoleon Nord (NapN) vent formed in 2014 [10,11,53,54]. La Soufriere's hydrothermal system appears to be distributed throughout all the volcanic edifice down to 2–3 km below the summit [11,50,52].

Degassing at La Soufriere was notably perturbed in February–April 2018, in concomitance with anomalous seismic activity (up to 180 earthquakes per day, in addition to four earthquakes felt by the population; [11]), which, on 27 April 2018, peaked at 4.1 M$_L$ (local magnitude), the strongest since the 1976–1977 eruptive period. According to changes in the fumarolic gas composition, this activity was interpreted as a failed phreatic eruption related to an anomalously heated and pressurized hydrothermal system that, in the same period, released a more "magmatic" signature [10,11]. Such a temperature–pressure increase in the hydrothermal system was accompanied by a nearly radial deformation pattern in the shallow part of the dome (according to Global Navigation Satellite System, GNSS, data) and the opening of fractures at rates of 3–7 mm per year [11,56].

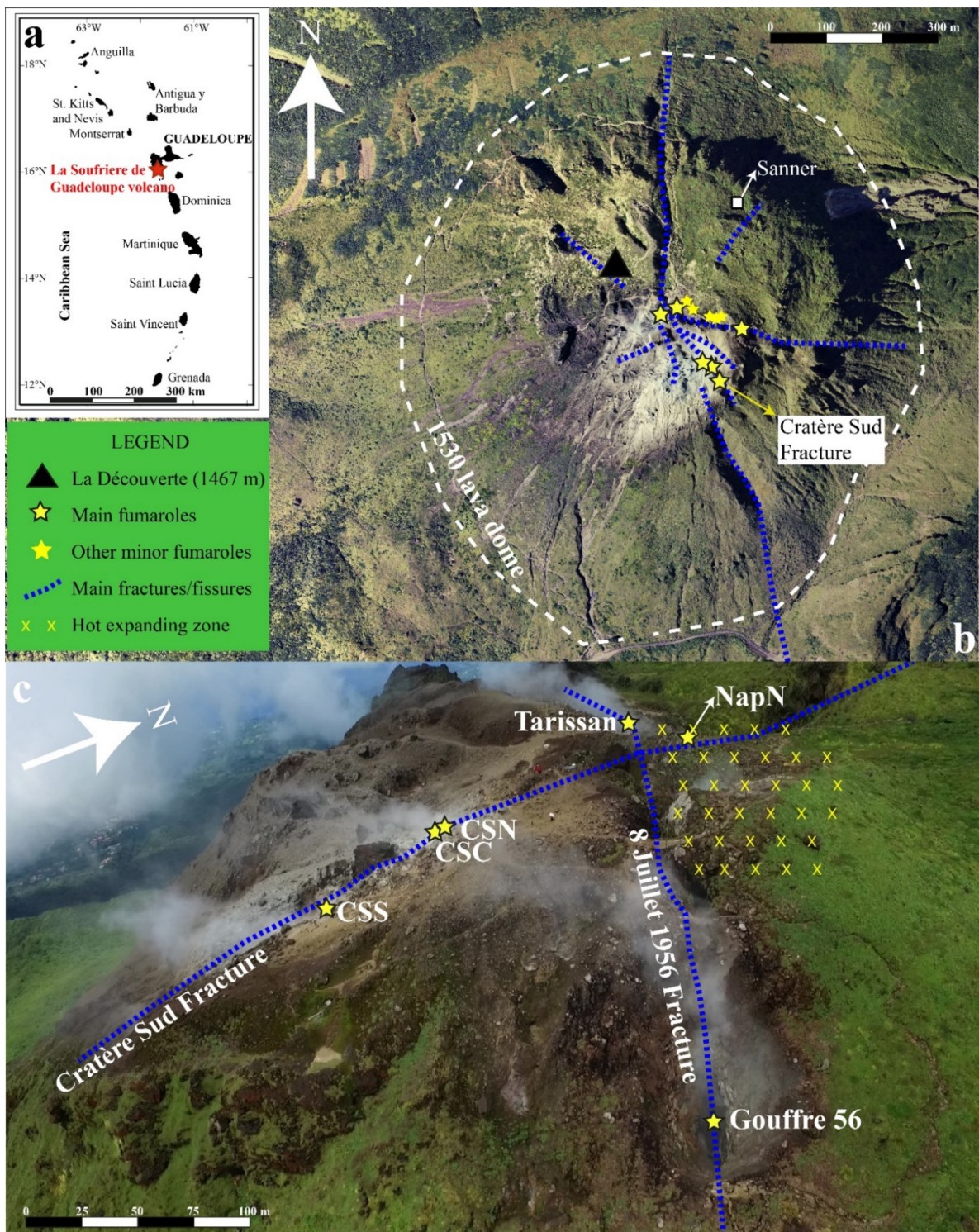

**Figure 1.** (**a**) Location map showing the position of Guadeloupe Islands regarding the Lesser Antilles volcanic arc. (**b**) Aerial photography of the 1530 dome (dashed white line) on the summit area of the La Soufriere de Guadeloupe volcano, including the distribution of fissures/fractures, main fumarolic emissions, and the position of the Sanner meteorological station. (**c**) Drone photography from the southeast portion of the dome showing the position of Cratère Sud and 8 Julliet 1956 fractures, the location of main fumarolic emissions (Cratère Sud Nord (CSN), Central (CSC), and Sud (CSS), plus Napoleon Nord (NapN) fumaroles), the Tarissan crater lake, and the Gouffre 56 pit, in addition to the ongoing hot expanding area (see text for further details).

The Cratère Sud fracture hosts Cratère Sud Sud (CSS), Cratère Sud Central (CSC) and Cratère Sud Nord (CSN) fumaroles. These vents are very close, with CSC and CSN being four meters from each other and representing two twin jet-type emissions (Figure 1c). The CSC fumarole was continuously monitored by the OVSG-IPGP, while CSN was sampled occasionally when accessible. After the 2018 volcanic unrest, CS fumaroles continued with fluctuating outlet gas temperatures (95 °C–111 °C [11,56]), variable chemistry of fumarolic gases [10,11], and visible morphological changes at the vent [57]. Temperature increases have been related to heat inputs in the roots of the hydrothermal system, while the temperature drops to the more significant input of meteoric waters and host-rock permeability changes [11,20]. Similarly, the chemistry of fumarolic gases displays variations during extended dry seasons, where an amplified magmatic signal of $SO_2$ arrives at the surface [10].

CS fumaroles were periodically sampled by the OVSG-IPGP, which has fully implemented the "Giggenbach" method [58] since mid-2017 (e.g., [11,59]). CS gases are, on average, made of 96–97 mol% of steam, 1.5–3 mol% of $CO_2$, <0.6 mol% $H_2S$, and minor $H_2$, $CH_4$, and CO. Sulfur dioxide, a typical magmatic-related compound, is absent in fumarolic fluids and can only be detected through multi-gas measurements at very low concentrations (e.g., $SO_2/H_2S$ ratios of ~0.05; [10,11,53]). These relative concentrations agree with those found in hydrothermal emissions and are considered to be due to the re-mobilized sulfur in the shallow portions of fumarolic systems, particularly soils around main emissions [10,53].

Recently, Inostroza et al. [60] investigated the chemical composition of condensates collected between 2017 and 2021. These authors determined that fumarolic emissions release significant amounts of metals and metalloids, as well as considerable amounts of lithophile and rock-related elements such as alkali and alkaline earths. The presence of metals and metalloids was primarily attributed to the input of a deep magmatic source and limited leaching from host rocks. At the same time, the presence of rock-related elements was ascribed to water–rock interaction processes. Here, we used the CS major and trace element data from the dataset published by Inostroza et al. [60], combined with other geochemical and geophysical data, to constrain the physicochemical processes within magmatic and hydrothermal systems.

## 3. Sampling and Analytical Methods

In this study, we used "Giggenbach" bottles and condensates collected by the OVSG-IPGP to determine major and trace gas components, as well as metals and metalloids released from fumarolic emissions at the volcanic vent. Fumarolic gases were collected in pre-evacuated ampoules (150 mL) filled with 50 mL of NaOH and Rotaflo® stopcocks. At the volcano summit, the ampoule was connected to a sampling line composed of a fiberglass tube, a Pyrex-Dewar glass elbow, and pipes joined with metallic clamps. Chemical analyses were carried out using the gas spectrometry technique (OmnistarTM analysis system) at the OVSG-IPGP. The analytical error was <10%. Condensates were sampled using a 50-cm-long fiberglass tube to channel the flow, connected to a Pyrex-Dewar glass elbow, following the sample collection procedure proposed by Chevrier and Le Guern [61] and explained in Inostroza et al. [60]. This procedure condenses fumarolic gas using Pyrex-Dewar pieces and cold water. Chemical analyses for trace elements were conducted through the Inductively Coupled Plasma Mass Spectrometry (ICP-MS, Agilent 7900) at the IPGP. Before chemical analyses, samples were filtered using syringes and a 0.20-μm-pore size sterile cellulose filters (VWR, number 514-0061). Internal standards for condensates samples (e.g., Sc, In, Re) and blank solutions were used to control possible error sources and validate results. Readers can refer to Moretti et al. [11,59] and Inostroza et al. [60] for details on sampling and analytical methods. The outlet gas temperature was recorded either at the end or at the beginning of gas + condensate sampling.

The groundwater level measure is helpful to determine fluctuations in meteoric water in the volcanic–hydrothermal system because we lack boreholes for direct measurements

of the water level. Here, we estimate the phreatic level height based on rainfall records by assuming that the amount of water added by steam condensation and the internal circulation of the hydrothermal system is negligible. This methodology is based on Darcy's law, where the drainage is proportional to the height of the groundwater table (e.g., [62]). The groundwater level variation at the time $t_i$ can be calculated as follows:

$$GWL(t_i) = GWL_0 - \sum_{n=0}^{i} \frac{p(t_n)}{\phi} e^{\frac{-(t_i - t_n)}{\tau}} \tag{1}$$

where $GWL_0$ is the asymptotic groundwater level, $\phi$ the effective porosity, $\tau$ the drainage time, and $p(t_n)$ the precipitation in the time $t_n$. Porosity and drainage data were taken for hydrothermally altered rocks in the presence of clay minerals, Fe-oxides, and silica polymorphs (e.g., [63,64]). In contrast, precipitation data were taken from weather stations located at the summit of the volcano and managed by the OVSG-IPGP, called Sanner (16.04497° N, 61.66272° W, 1411 m; Figure 1) and Savane à Mulets (16.038540° N, 61.665190° W, 1139 m). The groundwater level modeling was fitted with the Tarissan lake level, which was considered a direct indicator of the level of groundwaters making up the shallower section of the hydrothermal system. The absolute variation in the groundwater level was converted into the relative variation based on the average groundwater level in a specific year of reference (in this case, 2018), expressed as $GWL_y$, and the range of fluctuation between the minimum and maximum extreme valyes in the year of reference, expressed as $\Delta GWL_y$. The relative variation in the groundwater level, $dGWL/GWL$, is summarized as follows:

$$\frac{dGWL}{GWL}(t_i) = \frac{GWL(t_i) - GWL_y}{\Delta GWL_y} = \Delta GWL \tag{2}$$

Consequently, negative anomalies should be interpreted as a decreased groundwater levels related to 2018, while positive anomalies suggest a greater amount of groundwater within the hydrothermal system. However, given that the porosity and drainage data were the same for the entire period, these interpretations must be taken with caution, since $\Delta GWL$ is strongly dependent on the amount of rainfall and does not consider changes in the permeability of the rocks, a phenomenon observed in April 2018 by Moretti et al. [11].

## 4. Results

### 4.1. Giggenbach Bottles

We report on the gas molar ratios of $CO_2$, $CH_4$, and $S_T$ (Supplementary Material S1), which were contemporaneously collected with condensate samples [11,57,59], which help to determine the deep inputs of fluids coming from magmatic sources and fluctuations in the subsequent scrubbing processes.

The proportion of steam remained within the range of 95–98 mol.% (average 96.59 ± 0.18), except for the sample collected in June 2018, about 35 days after the peak in anomalous activity [11], where concentrations of 93.2 and 91.0 mol.% (92.1 mol% on average) of water vapor were recorded. $CO_2/CH_4$ ratios reached their maximum values in gas samples collected in April and August 2018, during the same period of the 2018 anomalous activity. Then, significant variations were noted in $CO_2/CH_4$ ratios, which gradually decreased until they reached minimum ratios in April 2020 and increased again, reaching higher values but lower than those recorded in 2018. Likewise, $CO_2/S_T$ ratios displayed values of ~3–6 during the entire study period, apart from the samples collected in October 2020, where ratios of ~8 were reached. These increments can be explained by the hot magmatic fluid inputs that increased the relative concentration of $CO_2$.

### 4.2. Major and Trace Element Concentrations in Gas Condensate Samples

Condensates were collected from the CS fumarole between April 2017 and January 2021 at a varying outlet temperature between 95.0 °C and 108.4 °C (Figure 2; Supplementary Material S1),

although the highest temperature recorded in the same period was 111 °C on 23 March 2018 (Figure 2). Almost all these samples were collected from the CSC vent. However, three samples (C8, C22, and C23; Supplementary Material S1) were collected from the CSN vent (the nearby "twin" fumarole). Despite some variable behavior occurring since 1992, these vents presented a similar performance during the study period according to their comparable chemical composition [10,53] and outlet gas temperatures [54]. Higher temperatures than the stable saturated steam vapor temperature were recorded at the summit (~95 °C) in March–April 2018, November 2019, and January 2021 at CS (Figure 2). It is notable that outlet gas temperatures were sporadically restored at 95 °C between 2017 and the first half of 2019, and temperatures were maintained above ~98 °C from the second half of 2019. Otherwise, the pH of condensates ranged from 0.92 to 4.52 (Supplementary Material S1), revealing greater acidity when temperatures reached their maximum values, especially since the second half 2019 (Figure 2).

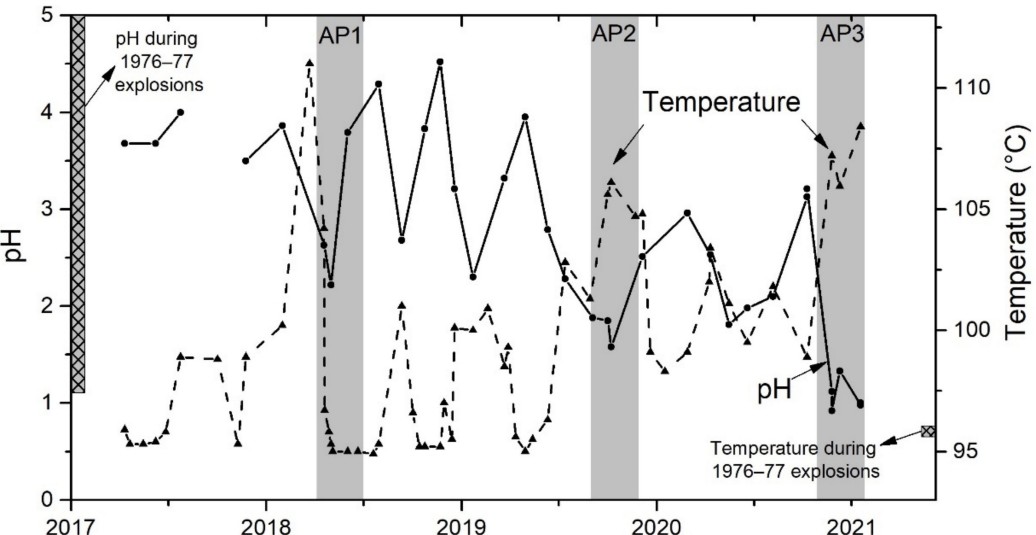

**Figure 2.** Outlet gas temperature (dashed black line) and pH (solid black line) of condensates from the CS fumarole for the 2017–2021 study period. Grey areas represent anomalous periods (AP) described in Section 4.2, while fill-pattern grey areas depict pH and temperature behavior during the 1976–1977 explosions [61]. Modified from Moretti et al. [11] and Inostroza et al. [60].

Chemical analyses presented low Relative Standard Deviation percentages (%RSD), ensuring data quality and representativity (Supplementary Material S1). RSD values were generally <20% (e.g., Li, B, Na, Mg, Al, Si, S, Cl, K, Ca, V, Cr, Mn, Fe, Co, Ni, Cu, Zn, Rb, Sr, Y, Mo, Ag, Sn, Sb, Cs, Ba, La, Ce, and Pb). In contrast, Tl and Bi presented medium–high RSD (>50%), plus Th and Be that reached RSD > 100% on average due to their near- or below-detection-limit concentrations. Condensates were predominantly composed of S, followed by Cl (concentrations of $10^1$ to $10^7$ ppb), two typical elements found in volcanic gases, in addition to Si ($10^2$ to $10^5$ ppb), an element leached from host rocks and linked with hydrothermal activity. The concentration of rock-forming chemical elements (Na, Mg, Al, Si, K, and Ca concentrations above $10^2$ ppb) was notably higher compared to chalcophile and siderophile elements (concentrations < $10^1$ ppb), which, in general terms, matched with worldwide (e.g., Costa Rican, Nicaraguan, Japanese volcanoes; [33,65]) condensate concentrations, but also suggests enhanced water–rock interaction during the study period. Water–rock interaction processes were also deduced following the behavior of halogens, which presented similar relative concentrations in host rocks [60] and condensates (i.e., higher concentrations of Cl and I compared to Br), suggesting incipient isochemical dissolution. This behavior was expected at La Soufriere volcano, where water–rock exchanges producing extensive hydrothermal alteration have already been recognized [11,42,53,55,64,66,67].

Concentrations of condensates randomly varied throughout the study period, generally with no clear tendencies. However, concentrations of Cl and B showed a relatively constant increment, along with the study period (Figure 3), reaching concentrations two and four orders of magnitude higher, respectively, than average concentrations prior to 2019. In contrast, Cr, Pb, and I mainly gradually increased from mid-2019 (Figure 3). Calcium, Ti, V, Al, and Br showed a slight decrease in their concentrations throughout the 2017–2021 period (Figure 3). Some chalcophile elements (e.g., Cu, Zn, and Sb; Figure 4) showed high concentrations during the first half of the study period, which were attributed to the anomalous activity caused by the input of heat and gases in the roots of the hydrothermal system in April 2018. However, most of the chalcophile elements behaved randomly.

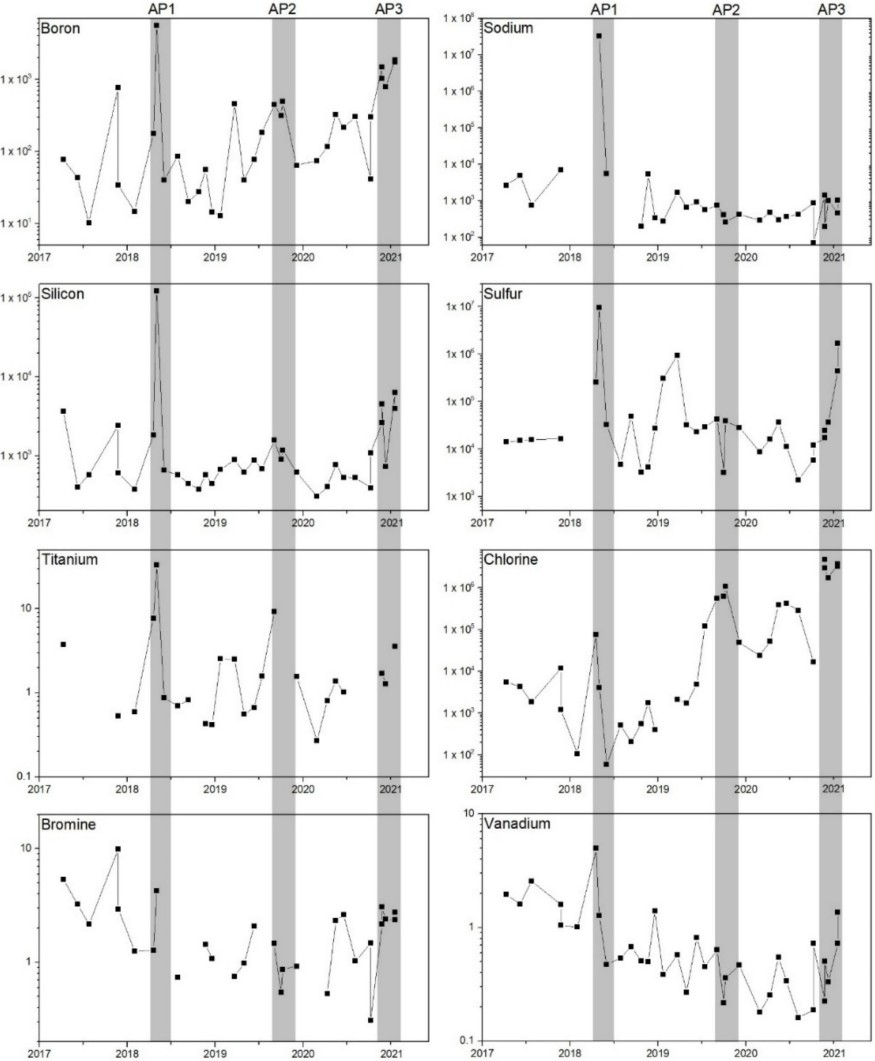

**Figure 3.** Concentration (in ppb) of selected major and trace elements in condensate samples at La Soufriere volcano. It is important to note that higher concentrations were reached within anomalous periods (grey areas described in Section 4.2). Missing data points (i.e., bromine) correspond to concentrations below the detection limit or high RSD > 50% (Supplementary Material S1).

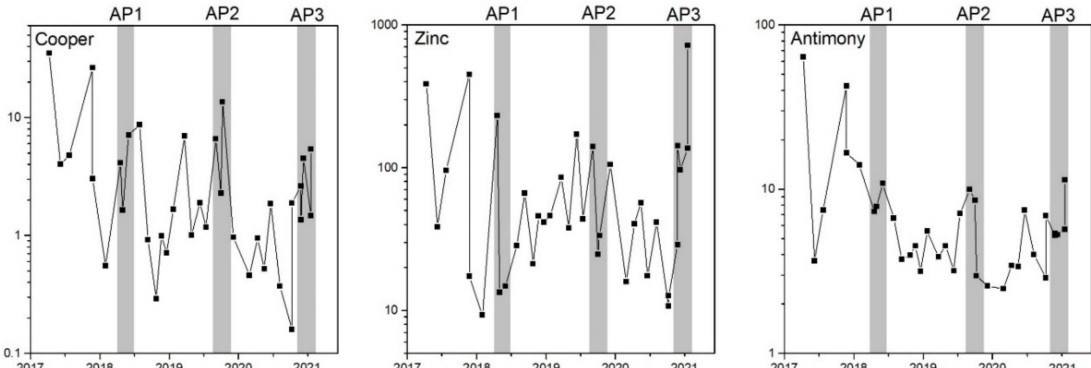

**Figure 4.** Variations in the concentration (in ppb) of representative chalcophile elements. Grey areas represent anomalous periods (AP) described in Section 4.2.

It is worth noting that steam proportion in fumarolic gases can affect the concentration of trace elements in the condensation process, given that higher steam proportions produce water-rich condensates, decreasing trace element concentrations [60]. Nevertheless, major and trace element concentrations associated with the sample collected in June 2018 (average steam of 92.1 mol%) were not affected by the small dilution effect (see Figures 3 and 4). Furthermore, the very high concentrations recorded in May 2018 (positive peaks in Figure 3) were reached when steam proportions were within average values (>95 mol%). Therefore, metal peaks can be directly linked to the internal and external processes governing the chemistry of fumarolic emissions.

### 4.3. Groundwater Level Variation

The ΔGWL ranged from −0.902 to 0.886. At the beginning of the studied period, maximum values were reached when rainfall frequently exceeded 400 mm per month (Figure 5). In contrast, minimum values were found in the second half of 2019 and mid-2020, which correlated with the extended dry seasons and monthly rainfall below 300 mm (Figure 5). Consequently, a tendency towards a more negative ΔGWL was observed as the study period elapsed (Figure 6), which agrees with the acidification of condensates and greater outlet gas temperatures (Figure 2).

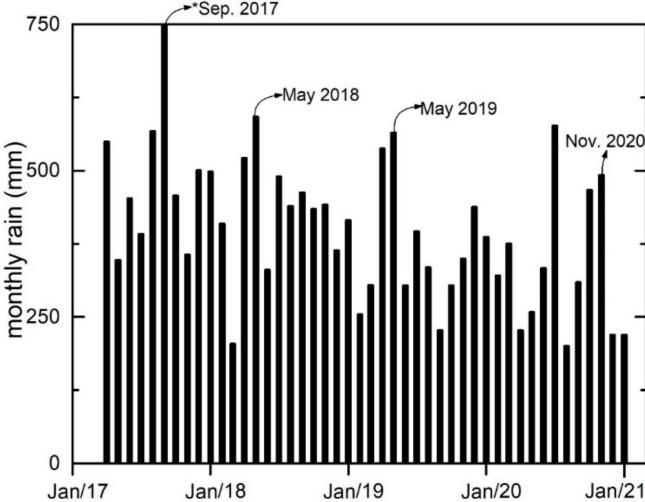

**Figure 5.** Monthly rain (in mm) measured by the Sanner weather station (operated by the OVSG-IPGP; 16.04497° N, 61.66272° W, 1076 m), which was located on the volcano summit. The data were extracted from monthly bulletins of the OVSG-IPGP [56]. * Monthly rainfall from September 2017 (>2000 mm) was overestimated because of technical issues in the weather station caused by Hurricane Maria.

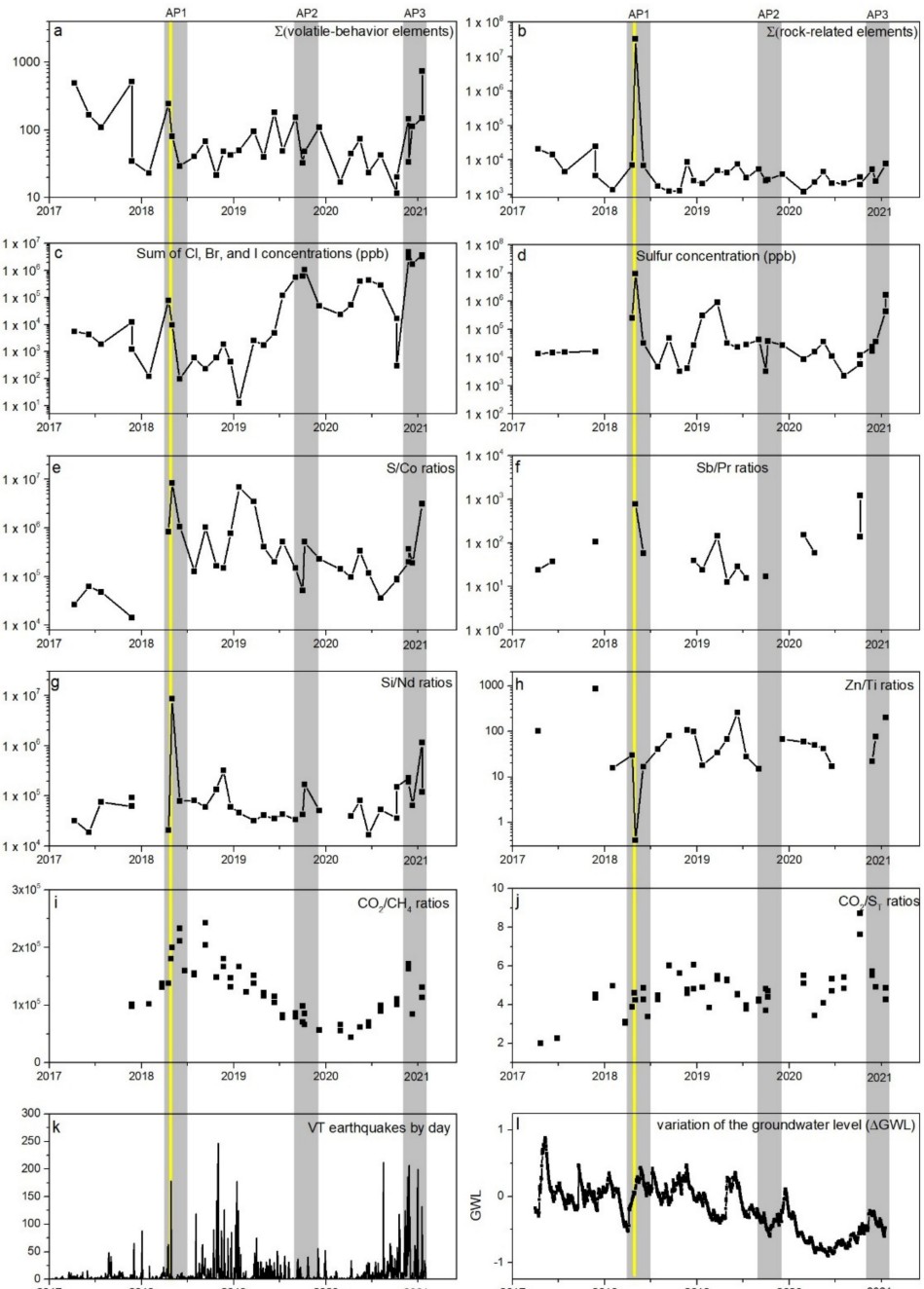

**Figure 6.** Time-series variations in major and trace elements in CS condensates linked to multiparametric data from the volcanic system and the CS fumarole (taken from bulletins of the OVSG-IPGP). (**a–d**) Variations in the concentration (in ppb) of volatile-behavior elements (S, Sb, B, Cl, Bi, As, Zn, Mo, Br, Cd, Ag, Ni, I, Pb, Tl, Sn, and Cu), rock-related elements (Be, Na, Mg, Al, Si, P, K, Ca, Ti, Cr, Mn, Fe, Rb, Sr, Y, Zr, Cs, Ba, REEs, Th, and U), the amount of easily scrubbed or highly soluble elements (Cl, Br, I), and the amount of S. (**e–h**) Variations in the S/Co, Sb/Pr, Si/Nd, and Zn/Ti ratios from condensate samples. (**i,j**) $CO_2/CH_4$ and $CO_2/S_T$ ratios from soda bottles collected monthly by the OVSG-IPGP and previously published by Moretti et al. [11]. (**k**) Number of volcano-tectonic (VT) earthquakes by day during the study period. (**l**) Variations in the groundwater level (ΔGWL), which rapidly responds to changes in the input of meteoric waters into the La Soufriere hydrothermal system. Gray areas depict anomalous periods (APs) described in Section 4.2, while the yellow line shows the felt earthquake $M_L$ 4.1 on 27 April 2018. Missing plots correspond to concentrations below the detection limit or high RSD > 50% (Supplementary Material S1).

### 4.4. Volcano–Tectonic Earthquakes

Unlike magma migration, the usual seismic activity in volcanic–hydrothermal systems such as La Soufriere reflects the hydraulic circulation of the two-phase convective hydrothermal body within the host rock and the perturbation of its stress regime by variable infiltrating bottom-up hot magmatic fluids and top-down phreatic groundwaters (e.g., [10,11,68,69]). Most seismic activity was clustered within seismic swarms, which occasionally reached more than 150 VT earthquakes per day (e.g., April and August–October 2018; [10,11]). Since January 2020, seismic activity has notably increased at La Soufriere [70], with a high number of very-low-energy VT earthquakes per day (Figure 6). Seismic activity concentrates in the shallow part of the volcanic edifice, at depths of 0.5–2 km below the volcano summit [10,11,56]. On the other hand, swarms reported prior to and during the April 2018 unrest located off the volcano axis had a hypocentral depth of 2–3.5 km below the summit and a higher intensity [10], peaking in the $M_L$ 4.1 event on 27 April 2018. This augmented activity can be related to the intensive rock fracturing triggered by the migration of hydrothermal fluids [11,56], comparable to the activity accounted for in April 2018.

## 5. Discussion

### 5.1. Definition of Anomalous Periods

The physicochemical characteristics of condensates allowed for us to determine three anomalous periods (hereafter called AP1, AP2, and AP3) based on two criteria: (i) increase in the Cl content and consequent decrease in the pH of samples, and (ii) increase in the concentration of certain trace elements (e.g., B, Fe, S, Pb, Si, Na; Figures 3 and 4; Supplementary Material S1). Although the outlet gas temperature had no direct influence on the chemical composition of fumarolic discharges in La Soufriere [60], higher temperatures at CS were usually reached during APs (Figure 2). The variations in these parameters in each AP are described below.

- AP1: This period includes the April–June 2018 samples collected during the 2018 unrest episode [11]. The outlet gas temperature reached 104.2 °C (e.g., sample C7) and 111.4 °C on 3 April 2018 [11], together with a noticeable pH decrease (down to 2.22; Figure 2) and higher Cl contents (Figure 3). In addition, the highest concentration in many chemical elements (e.g., Li, B, Na, Si, Ti, S, I, Ni, As, Sr; Figures 3 and 4) was recorded in AP1.

- AP2: This period includes samples collected between September and November 2019, where recorded temperatures reached 108.2 °C (November 2019). During AP2, the pH of samples was within the range of 1.58–2.51 (Figure 2), while Cl concentrations were higher than AP1 (Figure 3). Contrarily, trace element concentrations remained within the baseline values, except for Li, B, Cr, and Ag, which showed minor increments. Variations in the AP2 were recently ascribed to an extended dry season that reduced scrubbing processes in the hydrothermal system [10], which also explains the enhanced degassing of soluble species such as halogens.

- AP3: This includes samples collected between November 2020 and January 2021, where outlet gas temperatures reached 108.4 °C (January 2021). This period recorded the lowest pH (~1; Figure 2) and the highest Cl concentrations (Figure 3). In addition, various lithophile (Li, B, Si, Ti, I, Ba; Figure 3) and chalcophile (Cu, Zn, Sb, Cd, Pb, Bi; Figure 4) elements increased their concentrations during AP3.

### 5.2. Monitoring

According to Inostroza et al. [60], the origin of major and trace elements in La Soufriere's low-temperature emissions can be ascribed to the variable deep magmatic and shallow hydrothermal contributions. These processes were described until 2019, using the chemistry of fumarolic gases at La Soufriere [10,11]. However, there is no information in terms of major and trace elements. Therefore, to better understand these physicochemical processes, our major and trace element concentrations in condensates were correlated with seismological (number of events per month) and geochemical data ($CO_2/CH_4$ and $CO_2/S_T$ ratios) from the OVSG-IPGP, in addition to the ΔGWL.

Major and trace elements from condensates were divided into four groups according to their origin and the physicochemical processes that they suggest. First, volatile-behavior elements (S, Sb, B, Cl, Bi, As, Zn, Mo, Br, Cd, Ag, Ni, I, Pb, Tl, Sn, and Cu) are those elements transported within the gas and aerosol phases (e.g., [33,37,60]), whose origin is more related to a magmatic source, i.e., initially exsolved from a deep magmatic source, rather than a hydrothermal one. Second, rock-related elements (Na, Mg, Al, Si, P, K, Ca, Ti, Cr, Mn, Fe, Rb, Sr, Y, Zr, Cs, Ba, REEs, and U) correspond to rock-forming elements, which are usually found in high concentrations because of the leaching of andesitic rocks [60]. Third, halogen elements (Cl, Br, and I) are easily scrubbed or highly soluble in the presence of a hydrothermal system. Thus, their concentration variations can be used as an indirect indicator of the amount of water inside the volcano. Finally, the concentration of S, a chemical element of magmatic origin, is considered to trace the inputs of deep magmatic fluids.

During AP1 (Figures 6 and 7), volatile, rock-related, and soluble elements displayed positive peaks (Figure 6a–c), while the ΔGWL marked a negative value, which then returned to baseline (values, Figure 6l). AP1 was highlighted because of a prominent peak in rock-related elements (Figure 6b), which were mainly controlled by high concentrations of Na and Si (Figure 3). Such peaks suggested intense water–rock interactions related to the increased temperature-pressure conditions in the hydrothermal reservoir, which enhanced lithophile element removal from host rocks (e.g., [26]). This mechanism was also supported by high Si/Nd (Figure 6g) and low Zn/Ti (Figure 6h) ratios, which suggested higher concentrations of leachable Si and Ti in condensates of this period compared with non-volatile (i.e., Nd) and volatile (i.e., Zn) chemical elements. Remarkably enhanced water–rock interaction processes, overpressurization and heating of the hydrothermal system were also responsible for the anomalous seismic activity of April 2018 (Figure 6), which encompassed two seismic swarms concentrated on 16–17 April (140 VT earthquakes; one felt) and 27–28 April (180 VT earthquakes; two felt, including the $M_L$ 4.1). The input of heat and deep magmatic gases at the roots of the hydrothermal system can be supported by the higher S/Co and Sb/Pr ratios, which involve elements with the opposite volatile behavior at La Soufrière volcano [60], by assuming (i) the magmatic origin of S and Sb (the origin of Sb is still inconclusive, but its concentration is high in subduction-related magmatic-hydrothermal systems; [37,71,72] and (ii) a hydrothermal rock-related origin of Co and Pr. Consistent with this, the high $CO_2/CH_4$ ratios during the AP1 (Figure 6i) strongly point to the input of hot and $CO_2$-rich magmatic gases into the hydrothermal system [11,73].

During AP2 (Figures 6 and 7), volatile, rock-related, and S concentrations (Figure 6a,b,d) were within baseline values. However, the sum of Cl, Br, and I (Figure 6c) reached a positive peak, while ΔGWL showed low values. This variability points to the decreasing water/rock volume ratio of the hydrothermal reservoir (i.e., less water inside the volcanic system), which probably became a source of halogens being stripped into vapor from concentrated solutions [43,74]. Moune et al. [10] also noted an increase in the $SO_2/H_2S$ ratio (from 0.009 in March 2019 to 0.11 in October 2019), interpreted as an amplified magmatic signal, caused by a lower than normal extent of the scrubbing of deep-magma-derived gases. The faintly Si/Nd-positive peak (Figure 6g) could be ascribed to host-rock dissolution, given the more acidic nature of discharged fluids (pH < 2; Figure 2), rather than the heating and overpressurization of the hydrothermal aquifer, as also supported by the lower number of VT earthquakes (Figure 6k) during AP2. An increased input of magmatic gases, such as during AP1, was discarded due to the low S concentrations, low S/Co, Sb/Pr, and $CO_2/CH_4$ ratios (Figure 6).

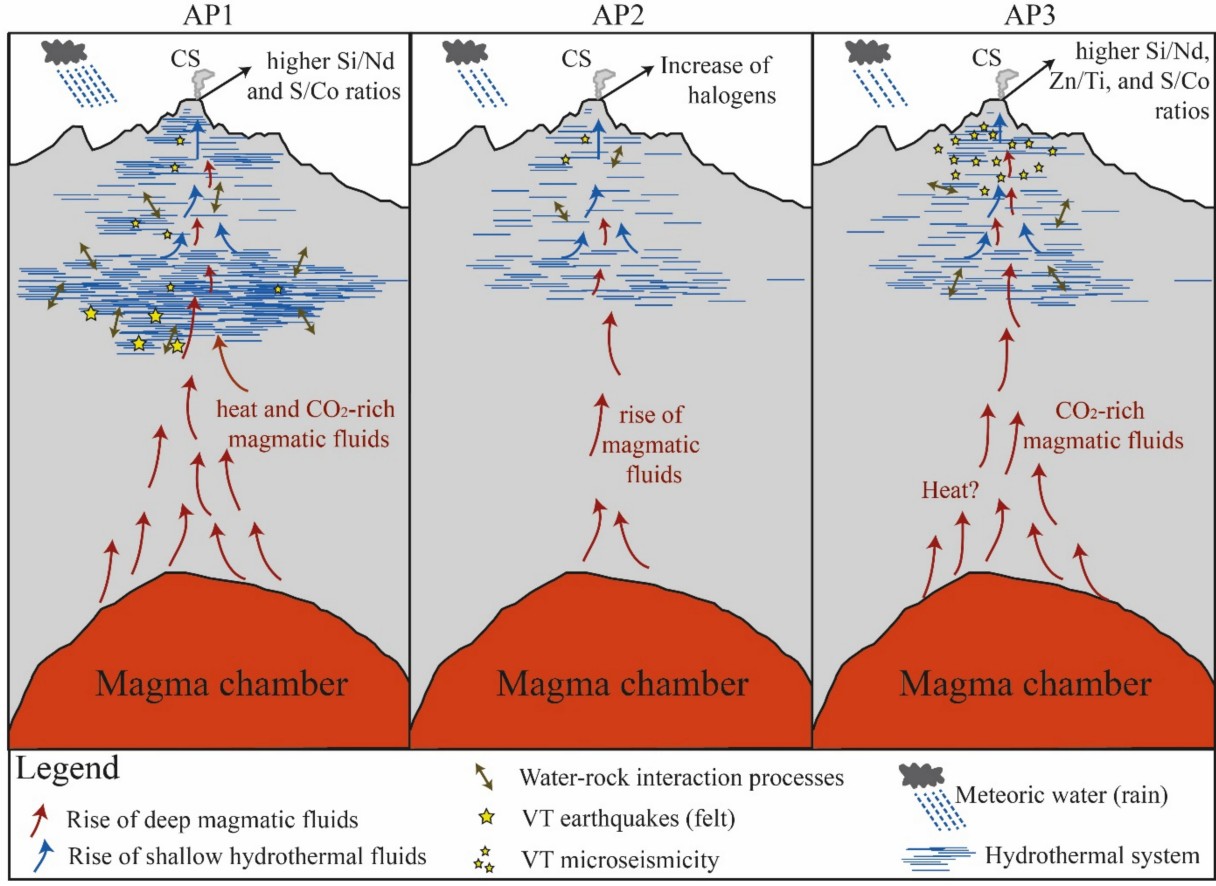

**Figure 7.** Sketch describing the three anomalous periods (AP1, AP2, and AP3). These conceptual models summarize the internal and external factors that produced APs, focusing on the input of deep magmatic gases, the water–rock interaction processes, the input of meteoric water (rainfall), and the amount of water within the hydrothermal system that enhances or decreases the magmatic signal of discharged fluids. Further details in the text. Figures are not to scale.

During AP3 (Figures 6 and 7), positive peaks were noted in volatile (Figure 6a), soluble elements (Figure 6c), and S concentrations (Figure 6d), whereas the ΔGWL remained close to the lower values (Figure 6l) of the 2017–2021 period. Although the hydrothermal reservoir was volumetrically diminished, similar to AP2, AP3 differed due to the input of deep magmatic gases revealed by the aforementioned positive peaks, plus the higher S/Co, Sb/Pr, and $CO_2/CH_4$ ratios (Figure 6e,f,i). We suggest that the infiltration of magmatic acid gases increased the temperature and pressure of the hydrothermal reservoir, as already observed at La Soufriere in 2018 [11]. This latter process favored the cracking of previously sealed regions (e.g., [75,76]), further stimulating the hydrothermal fluid circulation. This intensive rock fracturing explains the high number of VT earthquakes (Figure 6k) in the volcanic edifice. AP3 had similar features to AP1, but the extent of water–rock interaction due to the rapid heating and overpressurization stage was probably less accelerated than that during AP1, when VT earthquakes of a higher magnitude had a deeper but still hydrothermal origin (around 2.5 km below the summit [11]). A similar hypothesis could be sustained using the Zn/Ti ratio, where Zn represents a soluble and volatile-behavior element, and Ti is a typical hydrothermal product resulting from removal from the reservoir's host rocks after enhanced water–rock interaction [77,78]. The higher Zn/Ti ratios in AP3 than in AP1 (Figure 6h) suggest a weaker water–rock interaction that leached only limited Ti amounts from host rocks and favored higher contents of Zn, given the lower water/rock volume ratio in the hydrothermal system. Based on our condensate and multivariable data, presented in Figure 6, we speculate that during AP3, a deep pulse

of metal-bearing $CO_2$-rich gas was invested in the depleted hydrothermal system and enhanced hydrofracturing/hydroshearing. This promoted water–rock interaction without causing the strong pressure build-up and heating stage observed for AP1 [11].

The variable physicochemical behavior recorded from CS condensates allows for the different short-term processes during APs to be constrained in La Soufriere's hydrothermal system. However, at least four general trends occurred throughout the study period (Figures 2–6): (i) increase in outlet gas temperatures, (ii) acidification of condensates due to increments in Cl concentration, (iii) increase in $CO_2$ in fumarolic emissions, and (iv) increase in seismic activity. These trends evidence that the hydrothermal reservoir tends to decrease in size from the beginning of the study period, remarkably since mid-2019, when monthly rainfalls and ΔGWL decreased (Figure 5). Consequently, discharged fluids presented a less scrubbed signature at the end of the study period.

A decrease in the water/rock volume ratio of the hydrothermal reservoir can be explained by the minor recharge of meteoric waters (i.e., external force) given a more extended dry season (Figure 5), which seems to be a binding external force in volcanic–hydrothermal systems in a tropical environment. However, the effect of internal forces appears to be related to short-term (for a few months) pulsating perturbations in the hydrothermal system dynamics. Thus, an internal force cannot explain the long-term trends described above. Consequently, the hydrothermal activity and degassing processes at La Soufriere seem to primarily be regulated by the amount of rainfall feeding the hydrothermal aquifer. On the other hand, perturbations caused by the inputs of magmatic fluids had a short-term but considerable impact on fumarolic and hydrothermal activity.

## 6. Conclusions

This work reports the behavior of major and trace elements in fumarolic condensates collected between April 2017 and January 2021 from the Cratère Sud fumarole at La Soufriere de Guadeloupe volcano. Combining geochemical and seismic data allows for us to determine three anomalous periods during the study period. The first (April–June 2018) was caused by the input of deep magmatic gases in the root of the hydrothermal system, which increased the reservoir temperature–pressure, water–rock interactions, and element uptake into the vapor and aerosol phases. The second (September–November 2019) was characterized by above-average halogen emissions and decreased water/rock volume ratios, both caused by an extended dry season. Finally, the third (November 2020–January 2021) was triggered by the controlled input of magmatic gases into a relatively depleted hydrothermal system (low water/rock volume ratio), showing more decreased scavenging processes than the first anomalous period. Notable halogen increases (mainly Cl) during the second and third anomalous periods can be ascribed to the vaporization and stripping of the impoverished hydrothermal system. The decreased water/rock volume ratios of the hydrothermal system can be related to the remarkable minor inputs of meteoric water in 2019–2021, which suggest rainfall's (an external force) primary control of the functioning of the hydrothermal system and fumarolic activity.

Time-series analyses on condensate samples recorded physiochemical changes in the hydrothermal reservoir related to the variable interplay of deep (magmatic) and shallow (hydrothermal) factors, which are both critical parameters in the case of future magmatic or nonmagmatic eruptions. The sum of rock-related major and trace elements, volatile-behavior elements, and variations in the concentration of sulfur and chlorine, are proposed as suitable indicators of hydrothermal activity. Thus, monitoring these changes is helpful in constraining the physicochemical processes during volcanic–hydrothermal unrest.

**Supplementary Materials:** The following supporting information can be downloaded at: https://www.mdpi.com/article/10.3390/geosciences12070267/s1, Supplementary Material S1: trace element data processing and multiparametric data.

**Author Contributions:** Conceptualization, M.I., S.M., R.M. and M.B.; methodology, M.I., S.M., V.R., E.C.-E., P.B. and A.B.; formal analysis, M.I.; investigation, M.I., S.M. and R.M.; data curation, M.I., P.B. and A.B.; validation, M.I., S.M., V.R. and P.B.; writing—original draft, M.I.; writing—review and editing, M.I., S.M., R.M. and P.B.; visualization, M.I.; supervision, S.M.; project administration, S.M. All authors have read and agreed to the published version of the manuscript.

**Funding:** This work was supported by the EU-funded project IMMERGE (Impact multi-environnemental des retombées volcaniques et sahariennes en Guadeloupe; PI: Céline Dessert, IPGP). This project has been financially supported by European (FEDER FSE PO 2014–2020) and Région Guadeloupe funding (agreement GP0023419). R. M. acknowledges financial support from the project INSU-CNRS Tellus-Aléas "Unrest", year 2021. This study contributes to the IdEx Université de Paris ANR-18-IDEX-0001 and the Laboratory of excellence ClerVolc number 549.

**Data Availability Statement:** The data of this study are available in Supplementary Materials files.

**Acknowledgments:** We thank at the team of the Observatoire Volcanologique et Sismologique de la Guadeloupe (OVSG-IPGP), especially Tristan Didier, Thierry Kitou, and Sébastien Deroussi, for helping VR and ECE in the sampling of condensates. All the authors also thank the IPGP for their recurrent funding to the Observatoires Volcanologiques et Sismologiques (OVS-IPGP), the INSU-CNRS for funding provided by Service National d'Observation en Volcanologie (SNOV), and the Office de l'Eau Guadeloupe via the OVSG/IPGP-OE971 agreement. ICP-MS analyses were supported by IPGP multidisciplinary program PARI, and by Paris-IdF region SESAME Grant n° 12015908.

**Conflicts of Interest:** The authors declare no conflict of interest.

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
