# Peer review of "Monitoring Hydrothermal Activity Using Major and Trace Elements in Low-Temperature Fumarolic Condensates: The Case of La Soufriere de Guadeloupe Volcano"

_geosciences, doi:10.3390/geosciences12070267_

Round 1

Reviewer 1 Report

The manuscript of Manuel Inostroza et al., “Monitoring Hydrothermal Activity Using Major and Trace Elements in Low-Temperature Fumarolic Condensates, The Case of La Soufriere De Guadeloupe Volcano” provides carefully collected data on variation of trace element contents condensates during long term 2017-2021y. This data help for better  understanding processes  accounting for deeper magmatic and shallow hydrothermal sources. Evolution the volatile element contents (e.g., S, Sb, B, Cl, Bi, Zn, Mo, Br, Cd, Ag, Cu, and Pb) and the amount of leached rock-related elements (e.g., Na, Mg, Al, Si, P, K, Ca, Ti, Cr, Mn, Fe, Rb, Sr, Y, Cs, Ba, REEs, and U), and and alsothe concentration of Cl and S are applied for the monitoring of the hydrothermal systems. Variable inputs of deep magmatic components in the root of the hydrothermal system are recognized due to analyses of pH and element of fluctuation. I think, this is a well written, illustrated and insightful study that provides good basement for scientific discussion and citation in future. I have no critical remarks, which could improve the content of the nice manuscript. I recommend to accept the manuscript in present form.

Author Response

Reply to Reviewer #1 comments

In the manuscript ID: geosciences- 1779694, the authors analyze the fumarolic activity and associated process through the data reported by monitoring La Soufriere de Guadeloupe Volcano. This paper is interesting because it provides new insights into the fumarolic activity of an active volcano for four years. However, the manuscript requires very few modifications before being published. Most comments below are not criticisms to be addressed absolutely, but mere suggestions for improvement, and I trust the authors to know better than I which ones are valuable to follow.

Response: Thank you very much for accepting to review the manuscript and give valuable comments. Each of your comments, suggestions, and corrections are addressed below.

COMMENTS

  1. General

 A valuable paper. This paper contributes novel data and interpretations of hydrothermal activity from geochemical and geophysical monitoring. The paper is competently written and easy to read and has been carried out using appropriate techniques to arrive at the conclusions provided by the authors. 

The writing is clear, and the structure is satisfying, although minor errors require a thorough review by the authors.

 The authors establish a sequential relationship between time-series analyses and several vents. They synthesize information containing the sequential evolution of temperature and pH of the sites, as well as their geochemical evolution. They characterize the geochemical behaviour of the elements present in the condensates and relate each group of elements to a different evolutive context of the volcanic system. 

The graphics included are high quality and adequately accompany the authors' descriptions. In addition, they present a summary image with the model corresponding to the three anomalous periods.

The patterns extracted from the data are carefully handled.

Overall, the paper contributes to understanding the evolution of hydrothermal systems associated with quiescent volcanoes. The data provided by the authors contribute to a better understanding of present-day volcanoes and help understand some peculiarities of some extinct volcanic systems.

Response: We appreciate your comments. Thank you.

  1. Main Comments

My main comments on the paper:

Line 218: "Porosity and drainage data were taken for hydrothermally altered rocks in the presence of clay minerals, Fe-oxides, and silicification". Silicification is the introduction of, or replacement by silica, generally resulting in the formation of fine-grained quartz, chalcedony, or opal. It, therefore, involves a process of hydrothermal alteration. Here you mix minerals and hydrothermal processes. The authors indicate "silication"; however, they do not indicate which silica-rich phases are present; for example, they do not indicate argillitisation but clays minerals.

Response: We fully agree with your suggestion. The word silicification was replaced by silica polymorphs, an expression used in previous works done in La Soufriere de Guadeloupe (e.g., Salaün et al., 2011)

III. References

Authors are aware of the most relevant literature and use it appropriately.

  1. Typos and nitpicking

Typographical errors and nitpicking are included in the attachment. I also include some minor suggestions about the text.

Line 119: the word sub-plinian was corrected as suggested

Line 124: "volcano surroundings" was changed by "volcano's surroundings"

Line 129: the sentence was corrected as suggested

Line 157: the sentence was corrected as suggested

Line 218: this word was corrected as suggested

Reviewer 2 Report

In the manuscript ID: geosciences- 1779694, the authors analyze the fumarolic activity and associated process through the data reported by monitoring La Soufriere de Guadeloupe Volcano. This paper is interesting because it provides new insights into the fumarolic activity of an active volcano for four years. However, the manuscript requires very few modifications before being published. Most comments below are not criticisms to be addressed absolutely, but mere suggestions for improvement, and I trust the authors to know better than I which ones are valuable to follow.

COMMENTS

I. General

 A valuable paper. This paper contributes novel data and interpretations of hydrothermal activity from geochemical and geophysical monitoring. The paper is competently written and easy to read and has been carried out using appropriate techniques to arrive at the conclusions provided by the authors. 

The writing is clear, and the structure is satisfying, although minor errors require a thorough review by the authors.

 The authors establish a sequential relationship between time-series analyses and several vents. They synthesize information containing the sequential evolution of temperature and pH of the sites, as well as their geochemical evolution. They characterize the geochemical behaviour of the elements present in the condensates and relate each group of elements to a different evolutive context of the volcanic system. 

The graphics included are high quality and adequately accompany the authors' descriptions. In addition, they present a summary image with the model corresponding to the three anomalous periods.

The patterns extracted from the data are carefully handled. 

Overall, the paper contributes to understanding the evolution of hydrothermal systems associated with quiescent volcanoes. The data provided by the authors contribute to a better understanding of present-day volcanoes and help understand some peculiarities of some extinct volcanic systems.

II. Main Comments

My main comments on the paper: 

Line 218: "Porosity and drainage data were taken for hydrothermally altered rocks in the presence of clay minerals, Fe-oxides, and silicification". Silicification is the introduction of, or replacement by silica, generally resulting in the formation of fine-grained quartz, chalcedony, or opal. It, therefore, involves a process of hydrothermal alteration. Here you mix minerals and hydrothermal processes. The authors indicate "silication"; however, they do not indicate which silica-rich phases are present; for example, they do not indicate argillitisation but clays minerals. 

III. References

Authors are aware of the most relevant literature and use it appropriately. 

IV. Typos and nitpicking

Typographical errors and nitpicking are included in the attachment. I also include some minor suggestions about the text.

Author Response

(The authors gave the same response as above.)

Reviewer 3 Report

Dear editor and authors,
The current manuscript under consideration by Inostroza et al is an interesting record of chemical indicators over time in the La Soufriere volcano. I don't really have any major issues with the paper, except one:
You often talk about volatile elements that are coming from the magma, and elements that are coming from the hydrothermal activity and rock interaction. Some of the arguments are a bit circular: "How do we know there was hydrothermal activity? Because we see rock-like elements. How do we know they're not from the magma? There was hydrothermal activity". I hope you understand my point. I think that there should be more focus on how exactly we know which elements are coming from the rocks, and which from the magmas, and maybe add a figure or two emphasising this topic, potentially with correlation diagrams or tables.

Other than that, see minor comments, mostly about clarity, below:

line 15: "has become" - is it more challenging now than it was in the past?

line 58: The word "screen" is ambiguous here, I suggest removing it

line 59: "Besides" is not a good word to start a sentence with

line 66: why parentheses? say deep magmatic, shallow hydrothermal

line 68: Both hands or both sides, be consistent

line 70: why parentheses? just say direct, not (direct)
The text in general has an excessive use of parentheses. I strongly recommend to check each case if it is necessary, as they interfere with the flow of reading.

line 98: Why aren't they reliable?

line 130: Unclear what is due to interaction with meteoric waters. The "they evolve" or the "they ascend"?

line 167: Please clarify what a "hydrothermal stage" is

line 182: alkaline earths, not earth-alkalis

line 210: lack instead of miss?

line 211: height

line 213: negligible

line 215: subscript i?

line 237: A figure will be useful here to show variation of Giggenbach bottle results over time

line 255: 95 is two significant figures, and 108.4 is four, implying varying precision. Is that so?

line 279: How much is "mainly"? 50%? 99%?

line 281: How do you know that Si isn't coming, at least partly, from the volcanic gas?

line 282: Again, how do you know these elements aren't coming from the gas? Na and K, for example, are very volatile so I'd expect them in gas.

line 315: missing plots, or missing data points? The plots are all there, as far as I can see

fig 4: "Antimony", not "Antimonium"

line 332: "...caused by Hurricane Maria."

line 351: "Definition of anomalous periods"?

fig 7: Important element and element ratios used to infer the nature of the activity should be added to each panel in the figure

Author Response

Reply to Reviewer #2 comments

The current manuscript under consideration by Inostroza et al is an interesting record of chemical indicators over time in the La Soufriere volcano. I don't really have any major issues with the paper, except one:

You often talk about volatile elements that are coming from the magma, and elements that are coming from the hydrothermal activity and rock interaction. Some of the arguments are a bit circular: "How do we know there was hydrothermal activity? Because we see rock-like elements. How do we know they're not from the magma? There was hydrothermal activity". I hope you understand my point. I think that there should be more focus on how exactly we know which elements are coming from the rocks, and which from the magmas, and maybe add a figure or two emphasising this topic, potentially with correlation diagrams or tables.

Response: We appreciate the comments and suggestions of Reviewer 2, which are helpful in improving this work. Regrettably, trying to see magmatic contributions at La Soufriere volcano is challenging. Only volatile and less soluble chemical elements can be used as chemical markers for magma degassing. At La Soufriere, we infer magmatic inputs because of increments in the concentration of S and higher CO2/CH4 and CO2/St ratios, besides the presence of chalcophile elements (Sb, Bi, As, Zn, Cd, Ag, Pb, Tl, Sn, and Cu). This is because they present relatively high concentrations in the condensate regarding host rocks and are typically enriched in fumarolic emissions from magmatic-hydrothermal systems (Mandon et al., 2019; Aguilera et al., 2016; Inostroza et al., 2020). In the case of rock-related elements, we consider that determining their precise origin is outside the scope of this manuscript, given that specific isotopic analyses should be carried out, and even the origin of certain elements can remain inconclusive. In this manuscript, we focus more on the physicochemical processes that could lead to the enrichment of certain chemical elements instead of deciphering their origin.

Correlation diagrams using similar condensate samples were recently published by Inostroza et al., (2022). In this publication, determination coefficients among rock-relate elements are higher than those including Cl or S, which are usually used in magmatic degassing to better understand the transport of major and trace elements. These coefficients suggest that most rock-related elements found in condensates have a common source. In addition, Inostroza et al. (2022) presented chondrite-normalized REE patterns that match with those from andesitic rocks in La Soufriere. Consequently, the common source of rock-related elements in condensates can be only ascribed to the leaching of host rocks in the hydrothermal system. A magmatic source of rock-related elements can not be discarded, but if rock-related elements in ascending fluids reach the hydrothermal system, they should precipitate or dissolve; that is, they lose their magmatic imprint.

Given that correlation diagrams and discussion about the origin and processes causing fluctuations in major and trace elements were discussed by Inostroza et al. (2022), we do not incorporate them into this manuscript. Instead, we strengthen our interpretations by adding some extra references in Lines 392 and 396.

Other than that, see minor comments, mostly about clarity, below:

line 15: "has become" - is it more challenging now than it was in the past?

Response: "has become" was replaced by "is" to clarify the sentence.

line 58: The word "screen" is ambiguous here, I suggest removing it

Response: the word was removed as suggested

line 59: "Besides" is not a good word to start a sentence with

Response: The word was removed as suggested

line 66: why parentheses? say deep magmatic, shallow hydrothermal

Response: parentheses were removed as suggested

line 68: Both hands or both sides, be consistent

Response: the sentence was corrected as suggested

line 70: why parentheses? just say direct, not (direct)

The text in general has an excessive use of parentheses. I strongly recommend to check each case if it is necessary, as they interfere with the flow of reading.

Response: We revised the use of parentheses, and they considered less necessary were deleted, such as those found in Lines 70, 97-98, 227-228

line 98: Why aren't they reliable?

Response: the main problem with classic proxies involving sulfur in volcanic monitoring is related to the variable behavior of sulfur compounds in low-temperature hydrothermal emissions. In these emissions, i) SO2 is reduced to H2S, and ii) reactions among S-compounds can form sublimates and incrustations in the fumarolic conduit and at the surface. Also, we do not know precisely how fast and where H2S can be oxidized to form SO2 again… In low-temperature emissions, these processes are poorly constrained. Consequently, such sulfur partitioning in solid, liquid, and gas phases causes critical uncertainties in monitoring volcanoes such as La Soufriere de Guadeloupe. In the new manuscript version, a brief explanation of this subject is included.

line 130: Unclear what is due to interaction with meteoric waters. The "they evolve" or the "they ascend"?

Response: "ascend" was removed to avoid misunderstood

line 167: Please clarify what a "hydrothermal stage" is

Response: Hydrothermal stage means those volcanic emissions where H2S is the predominant S specie and SO2 is in very low concentrations or below the detection limit. The presence of H2S is a direct indicator of hydrothermal activity, as stated by several authors in the last decades (Giggenbach 1980; Giggenbach 1996; Fischer and Chiodini 2015). The word "hydrothermal stage" was replaced by "hydrothermal emissions" to avoid confusion

line 182: alkaline earths, not earth-alkalis

Response: the sentence was corrected as suggested

line 210: lack instead of miss?

Response: We chose "lack" instead "miss".

line 211: height

Response: the word was replaced as suggested.

line 213: negligible

Response: the word was replaced as suggested.

line 215: subscript i?

Response: "i" was subscripted as suggested.

line 237: A figure will be useful here to show variation of Giggenbach bottle results over time

Response: Figure 6 is already showing the results of Giggenbach bottles, and we believe that a new figure is unnecessary.

line 255: 95 is two significant figures, and 108.4 is four, implying varying precision. Is that so?

Response: We do not know what the meaning of your comment is. Temperatures were measured with the same device, and no varying precision was expected. We include one decimal (95.0 °C) in the minimum temperature at Cratère Sud fumarole, so minimum and maximum temperatures are in the same format.

line 279: How much is "mainly"? 50%? 99%?

Response: S, Cl, and Si concentrations accounted for more than 50% of the composition in condensates. The word "mainly" was replaced by "predominantly (>50%)".

line 281: How do you know that Si isn't coming, at least partly, from the volcanic gas?

Response: coming from gas, are you referring to it coming from magma degassing? In low-temperature hydrothermal emissions, the presence of Si is well correlated with leaching processes from host rocks. Si inputs from deeper portions in the magmatic system could be plausible, but precipitation and dissolution processes begin as soon they contact the hydrothermal system. We do not discuss deeper inputs of Si and other rock-related elements because characteristics of the magmatic inputs are obscure by the hydrothermal system, and only the dry gas chemistry of insoluble species and a few volatile-behavior elements found in condensates can give insights into the magmatic degassing. Therefore, independently of the behavior of Si beneath hydrothermal reservoirs, we interpret their fluctuations as water-rock interaction processes caused by temperature and pressure variations. This idea was taken from geothermometric studies done in geothermal areas, which use Si to calculate the temperature reservoir, being its solubility higher as reservoir temperature-pressure increases (e.g., Fournier 1973, 1991; Arnorsson 1975; Fournier and Potter 1982).

line 282: Again, how do you know these elements aren't coming from the gas? Na and K, for example, are very volatile so I'd expect them in gas.

Response: Na, K, and, in general terms, most rock-related elements are poorly volatile in low-temperature volcanic emissions because they prefer partitioning into the fluid phase (e.g., Pokrovski et al. 2013), and they are only found in the hotter section of temperature-dependent tests in silica tubes (e.g., Africano et al. 2002; Yudovskaya et al. 2006). Their high concentration in condensates is due to water-rock interaction related to hydrothermal activity, as mentioned above and in section 5.2 (i.e., Line 395). The scope of this manuscript is not determining the origin of major and trace elements because specific studies are necessary for that. Instead, this manuscript state that their concentration in condensate reflects their solubilities in the hydrothermal reservoir, which depend on temperature-pressure conditions. Consequently, if there is even a minimal magmatic contribution of rock-related elements, they should dissolve or precipitate into the hydrothermal reservoir, which immediately converts them into hydrothermal tracers.

line 315: missing plots, or missing data points? The plots are all there, as far as I can see

Response: the sentence was modified as suggested

fig 4: "Antimony", not "Antimonium"

Response: the word was corrected in figure 4 as suggested

line 332: "...caused by Hurricane Maria."

Response: The sentence was modified as suggested

line 351: "Definition of anomalous periods"?

Response: The word was corrected as suggested

fig 7: Important element and element ratios used to infer the nature of the activity should be added to each panel in the figure

Response: Figure 7 was modified as suggested.
